# Utilization Trend and Comparison of Different Radiotherapy Modes for Patients with Early-Stage High-Intermediate-Risk Endometrial Cancer: A Real-World, Multi-Institutional Study

**DOI:** 10.3390/cancers14205129

**Published:** 2022-10-19

**Authors:** Kang Ren, Lijuan Zou, Tiejun Wang, Zi Liu, Jianli He, Xiaoge Sun, Wei Zhong, Fengju Zhao, Xiaomei Li, Sha Li, Hong Zhu, Zhanshu Ma, Shuai Sun, Wenhui Wang, Ke Hu, Fuquan Zhang, Xiaorong Hou, Lichun Wei

**Affiliations:** 1Department of Radiation Oncology, Peking Union Medical College Hospital, Chinese Academy of Medical Sciences & Peking Union Medical College, Beijing 100730, China; 2Department of Radiation Oncology, The Second Hospital of Dalian Medical University, Dalian 116023, China; 3Department of Radiation Oncology, The second hospital Affiliated by Jilin University, Changchun 130041, China; 4Department of Radiation Oncology, First Affiliated Hospital of Xi’an Jiaotong University, Xi’an 710061, China; 5Department of Radiation Oncology, The General Hospital of Ningxia Medical University, Yinchuan 750003, China; 6Department of Radiation Oncology, The Affiliated Hospital of Inner Mongolia Medical University, Hohhot 750306, China; 7Gynaecological Oncology Radiotherapy, Affiliated Tumor Hospital, Xinjiang Medical University, Urumqi 830054, China; 8Department of Radiation Oncology, Gansu Provincial Cancer Hospital, Lanzhou 730050, China; 9Department of Radiation Oncology, Peking University First Hospital, Beijing 100034, China; 10Department of Radiation Oncology, The 940th Hospital of Joint Logistics Support force of Chinese People’s Liberation Army, Lanzhou 730050, China; 11Department of Radiation Oncology, Xiangya Hospital Central South University, Changsha 410008, China; 12Department of Radiation Oncology, Affiliated Hospital of Chi feng University, Chifeng 024050, China; 13State Key Laboratory of Complex Severe and Rare Diseases, Peking Union Medical College Hospital Chinese Academy of Medical Sciences & Peking Union Medical College, Beijing 100730, China; 14Department of Radiation Oncology, Xijing Hospital, Air Force Medical University of PLA (the Fourth Military Medical University), Xi’an 710068, China

**Keywords:** high–intermediate-risk, endometrial cancer, trend analysis, adjuvant radiotherapy, propensity score matching

## Abstract

**Simple Summary:**

The adjuvant treatment for early-stage endometrial cancer (EC) has remained an intractable problem in clinical practice. Although several risk classification strategies have been proposed to guide precise treatment, a significant proportion of patients are still overtreated, especially patients with high-intermediate-risk (HIR) early-stage EC. Here, we compared the survival outcomes between different adjuvant radiotherapy modes in patients with HIR EC defined by three primarily-used criteria, based on multicenter data, to provide further evidence for the adjuvant treatment choices for HIR patients. This study revealed multicentric utilization trends for different radiotherapy (RT) modes for the first time. It confirmed that pelvic external beam radiation therapy (EBRT) showed a survival advantage over vaginal brachytherapy (VBT) alone only in selected patients with HIR.

**Abstract:**

This study aimed to compare the outcomes of RT modalities among patients who met different HIR criteria based on multicentric real-world data over 15 years. The enrolled patients, who were diagnosed with FIGO I-II EC from 13 medical institutes and treated with hysterectomy and RT, were reclassified into HIR groups according to the criteria of GOG-249, PORTEC-2, and ESTRO-ESMO-ESGO, respectively. The trends in RT modes utilization were reviewed using the Man-Kendall test. The rate of VBT alone increased from zero in 2005 to 50% in 2015, which showed a significant upward trend (*p* < 0.05), while the rate of EBRT + VBT utilization declined from 87.5% to around 25% from 2005 to 2015 (*p* > 0.05). There were no significant differences in OS, DFS, LRFS, and DMFS between VBT alone and EBRT ± VBT in three HIR cohorts. Subgroup analyses in the GOG-249 HIR cohort showed that EBRT ± VBT had higher 5-year DFS, DMFS, and LRFS than VBT alone for patients without lymph node dissection (*p* < 0.05). Thus, VBT could be regarded as a standard adjuvant radiation modality for HIR patients. EBRT should be administrated to selected HIR patients who meet the GOG-249 criteria and did not undergo lymph node dissection.

## 1. Introduction

Endometrial cancer (EC) is one of the most commonly diagnosed gynecological cancers worldwide [1]. In China, the estimated number of new patients diagnosed with EC was 81,964 in 2020, accounting for 19.64% of global EC incidence [2]. Among the newly diagnosed EC patients, approximately 80% of them were diagnosed as stage I or II [3]. Adjuvant treatment decision-making is based on risk classifications to maximize the treatment efficacy and reduce overtreatment. Early-stage EC patients are classified into low-risk, intermediate-risk, and high-risk groups according to risk factors, including age, stage, depth of MMI, and the status of lymph-vascular space invasion (LVSI) [4,5]. GOG (Gynecologic Oncology Group) and PORTEC (Postoperative Radiotherapy for Endometrial Cancer) trials had further identified an additional subset of patients with a higher risk of recurrence, defined as the HIR group, from the original intermediate-risk group, and concluded that this subgroup of patients would benefit from RT [6,7].

Adjuvant RT modalities included vaginal brachytherapy (VBT), pelvic external beam radiation therapy (EBRT), and a combination of the two. Notably, the criteria for HIR vary from consensus to consensus and organization to organization, and different standards refer to various risk factors. Specifically, the latest ESMO-ESGO-ESTRO consensus conference guidelines released in 2021 refined the HIR group and defined stage I grade (G) 3 and stage II as the HIR group, which is quite different from the ESMO-ESGO-ESTRO standard of 2016 [8]. In addition, advanced age is considered one of the high-risk factors for EC, which has been taken into consideration by PORTEC-based risk grouping and GOG-249, but not by the ESMO-ESGO-ESTRO consensus. Other operative-related factors, such as the preformation of the lymph node resection, which were not included in the classification criteria, may help to choose patients who may benefit from RT. The same patient can be classified into different risk classifications according to these criteria and receive heterogeneous treatment recommendations. Consequently, how to recommend appropriate adjuvant treatment modes for HIR patients defined by different classification standards is still an intractable clinical problem.

Previous research has not agreed on the optimal adjuvant radiotherapy options. The GOG-249 failed to demonstrate the superiority of VBT plus chemotherapy over EBRT alone concerning overall survival [9]. In addition, PORTEC-2 reported reduced locoregional relapse rates in patients who received EBRT versus VBT. Still, patients who received EBRT had higher rates of gastrointestinal toxicity and secondary malignancies with no survival benefit [10]. Based on the ESMO-ESGO-ESTRO guideline (version 2016), patients with unequivocally positive LVSI and no surgical nodal staging are recommended for EBRT [5]. While for patients with IA G3 or IB G2, pelvic EBRT will be advised as category 2B evidence according to NCCN guidelines [11].

In current clinical practice, HIR patients tend to be administered de-escalation treatment. That is, VBT alone is preferred. However, a certain proportion of HIR patients are overtreated with EBRT or combined EBRT and VBT based on existing risk classifications. There are significant differences in health-economic burden and toxicities between the two RT modalities. Pelvic EBRT is delivered with a total dose of 45 to 50.4 Gy and takes about six weeks, while VBT only takes about two weeks with a total dose of 30 Gy [12].

Given that the radiotherapy choices for patients with HIR remain challenging, there have been no studies investigating how radiotherapy modalities evolved in China over a relatively long observation period, or examining how clinical trials have impacted the choice of radiotherapy for HIR early-stage EC.

Thus, the present study reviewed RT patterns for patients meeting the eligibility criteria for HIR. We aimed to analyze the survival benefit and toxicities between different radiotherapy modalities based on real-world multicenter data to provide further evidence for the adjuvant treatment choices for HIR patients.

## 2. Methods

### 2.1. Patient Eligibility Criteria

Patients with stage I to II EC at 13 grade A tertiary hospitals in China between Jan. 2000 and Dec. 2015 were retrospectively identified. All enrolled patients underwent adjuvant radiotherapy after a hysterectomy. Patients’ stages were converted to the 2009 FIGO staging system. Patients with the following clinical scenarios were excluded: follow-up period of fewer than three months and incomplete survival information. The study was approved by the Institutional Review Board of Peking Union Medical College Hospital (N0. S-K139) and has been registered in the Chinese clinical trial registry (registration number ChiCTR-PRC-17010712).

### 2.2. Treatment

Primary surgery included a total hysterectomy and bilateral salpingo-oophorectomy with or without lymphadenectomy. Patients who only received preoperative imaging assessment of the lymph nodes without lymphadenectomy staging were classified into the cN0 group, which was considered inadequate lymph node assessment. Patients who underwent lymphadenectomy or sentinel lymph node biopsy were categorized into the pN0 group after negative pathological confirmation, which was considered to be an adequate assessment. Adjuvant RT was administrated to all of the enrolled patients. RT included vaginal brachytherapy (VBT) alone, pelvic EBRT, or a combination of VBT and EBRT. EBRT, including four field box technique, IMRT, or VMAT, was delivered to the pelvic area at a total dose of 45–50.4 Gy in 23–28 fractions. VBT was delivered with a vaginal cylinder to the upper half of the vagina or 3–5 cm of the upper vagina. Patients who received VBT alone received 4–5 Gy per fraction in 5–8 fractions, and those who received a VBT boost after the completion of EBRT were administered doses of 5 Gy per fraction in 2–4 fractions. Intravenous concurrent or sequential adjuvant chemotherapy consisting of carboplatin/paclitaxel, cisplatin/doxorubicin, or cisplatin/doxorubicin/paclitaxel was administrated at the physician’s discretion.

Early radiation toxicities were evaluated by the Common Terminology Criteria for Adverse Events (CTCAE) version 3.0, and the late radiation toxicities were assessed by the Radiation Therapy Oncology Group criteria (RTOG).

### 2.3. Data Analysis

Overall survival (OS) was defined as the time from surgery to the date of death from any cause or last follow-up. Disease-free survival (DFS) was defined as the time from surgery to the date of treatment failure or death from any cause or last follow-up. LRFS (local recurrence-free survival) was defined as the time from surgery to the date of locoregional failure or death from any cause or last follow-up. DMFS (distant metastasis-free survival) was defined as the time from surgery to the date of distant metastasis or death from any cause or last follow-up.

Survival analysis was performed using SPSS statistical software (version 25.0; SPSS Inc., Chicago, IL, USA). The Kaplan-Meier method was applied to calculate survival data, and the log-rank test determined differences between groups. A *p*-value of <0.05 was considered statistically significant. A propensity 1:1 nearest neighbor-matching analysis (PSM) was performed to control the effects of potential confounders between VBT and EBRT ± VBT groups. The propensity score model included the following variables: age, MMI, LVSI, tumor grade, and type of surgery. The propensity score was obtained by logistic regression, and the match tolerance (caliper) was set at 0.1.

The Man-Kendall trend test was applied to assess whether there was an increasing or decreasing trend in the proportion of cases over the years. *p*-values < 0.05 were defined as statistically significant.

## 3. Results

### 3.1. Patients

Between January 1999 and December 2015, 1268 patients with early-stage EC from 13 Chinese medical institutions were enrolled. The basic information of the enrolled patients is shown in Table 1. The median follow-up time was 58 months (ranging from 6 to 237 months). Among patients receiving VBT alone, 5 Gy in six fractions was the most frequently used fractionation, while 5 Gy in two fractions was the most used VBT as a boost after EBRT. All enrolled patients were reviewed and further reclassified into HIR groups according to the GOG-249, PORTEC 2, and ESMO-ESGO-ESTRO criteria. Among the enrolled patients, 473 were identified as HIR according to the GOG-249 criteria, 184 were identified as HIR according to the PORTEC 2 criteria, and 204 met the ESMO-ESGO-ESTRO HIR criteria (Figure 1). In the GOG-249 cohort, 31.1% of patients received VBT and 68.9% received EBRT ± VBT. In the PORTEC-2 cohort, 51.6% of patients underwent VBT and 48.4% experienced EBRT ± VBT. Half of the patients who met the ESMO-ESGO-ESTRO criteria received VBT, and the other half received EBRT ± VBT.

### 3.2. The Man-Kendall Trend Analysis

Among all of the patients across the 13 institutions, VBT alone (44.6%) was the most common adjuvant RT modality, followed by EBRT plus VBT (34.7%) and EBRT alone (20.7%) during the whole follow-up. The proportion of women who received VBT alone significantly increased from 0% to 50% from 2002 to 2015 (*p* < 0.05). Whereas the rate of patients who underwent combined EBRT plus VBT declined from 87.5% to around 25% from 2005 to 2015; this trend did not reach statistical significance (*p* > 0.05). The rate of EBRT alone showed a decreasing trend (100% to 0%) from 2000 to 2005 and then slowly raised to a level of about 25% from 2005 to 2015 (*p* > 0.05).

As for patients who met the HIR criteria of GOG-249, the rate of VBT showed a flat-rise from 0% in 2002 to 37.5% in 2015, and this increasing trend was statistically significant (*p* < 0.001). In addition, the proportion of patients receiving EBRT showed a downward trend in the first five years (nearly 100% to 0%) and grew to 25% by 2015. The rate of combined EBRT and VBT increased from zero in 2002 to 87.5% in 2015 and slowly declined to 37.5% in 2015.

Meanwhile, for the HIR patients who met the PORTEC-2 criteria, the rate of VBT alone remained at zero before 2005 and then rose to 62.5% in 2015. The proportion of EBRT dropped from nearly 100% to 0% before 2005 and increased to 12.5% in 2015. The utilization trend of EBRT plus VBT began to decline from almost 100% in 2005 to around 17.5% in 2015. For the HIR group that met the ESMO-ESGO-ESTRO criteria, the RT mode utilization showed similar trends but in different proportions over the years. The rate of VBT alone rose from zero to over 50% in 2015. While the rate of EBRT ± VBT initially experienced a distinct increase from zero to nearly 100% in 2005, and subsequently declined to 37.5% in 2015, and the proportion of EBRT alone dropped from approximately 87.5% to zero between 2001 and 2006 and steadily rose to 12.5% in 2015 (Figure 2).

### 3.3. Survival Analyses in the Three Cohorts

In the GOG-249 cohort, the 5-year OS, 5-year DFS, 5-year DMFS, and 5-year LRFS rates were 93.30%, 84.20%,87.90%, and 89.00%, respectively, for patients who received VBT. For patients who received EBRT ± VBT, the corresponding statistics were 91.50%, 87.40%, 87.70%, and 92.20%, respectively. Survival outcomes showed no statistical differences even after propensity score matching with other factors, including age, surgery status, MMI, tumor grade, and LVSI, between the VBT alone and EBRT ± VBT groups (Table 2, Figure 3A–D).

In the PORTEC-2 cohort, the 5-year OS, DFS, DMFS, and LRFS rates were 96.00%, 93.50%, 92.50%, and 91.10%, respectively, for patients who received VBT. The corresponding statistics were 91.90%, 88.70%, 93.50%, and 94.50%, respectively, for EBRT ± VBT. There were no significant differences between the radiation groups before or after propensity score matching (*p* > 0.05) (Table 2, Figure 3E–H).

We performed a similar analysis in the ESMO-ESGO-ESTRO group. The 5-year OS, DFS, DMFS, and LRFS rates were 96.90%, 87.40%, 90.60%, and 89.20%, respectively, for the VBT group. The corresponding outcomes were 93.10%, 90.60%, 91.20% and, 94.60%, respectively, for the EBRT ± VBT group (*p* > 0.05) (Table 2, Figure 3I–L).

### 3.4. Subgroup Analyses in the GOG-249 HIR Cohort

Since a crossing of the survival curves was observed in the GOG-249 cohort, further subgroup analyses were performed according to surgery status and FIGO stage. For the cN0 patients’ group, the 5-year DFS, 5-year DMFS, and 5-year LRFS rates of patients who underwent EBRT ± VBT were significantly higher than patients who received VBT alone (*p* < 0.05). After 1:1 propensity-score matching, the two groups showed significant differences in DFS and LRFS (*p* < 0.05). While for the pN0 patients’ group, there were no significant differences in survival outcomes between EBRT ± VBT and VBT groups before and after matching (*p* > 0.05).

Different from the other three groups, GOG-249 also enrolled patients with FIGO II. Thus, we excluded this group of patients to perform the subgroup analysis. For patients who did not receive lymphadenectomy, EBRT ± VBT showed higher DFS than VBT alone (88.1% vs. 62.3%, *p* = 0.025). EBRT still had higher DFS and LRFS than VBT alone after matching (*p* = 0.049, and 0.044, respectively) (Appendix A, Figure 4).

### 3.5. Toxicities

After reclassifying all enrolled patients into HIR groups via the GOG-249, PORTEC 2, and ESMO-ESGO-ESTRO criteria, patients who underwent EBRT ± VBT experienced a higher rate of acute toxicities, including hematological, gastrointestinal, and urinary tract reactions, than the VBT group (*p* < 0.05) (Appendix A).

## 4. Discussion

The present study reviewed the radiotherapy modalities for patients eligible for the HIR criteria of GOG-249, PORTEC-2, or ESMO-ESGO-ESTRO in 13 Chinese medical institutions. Patients who received EBRT ± VBT had a significantly higher rate of acute radiation-induced toxicities. Although a considerable proportion of HIR patients still received EBRT ± VBT in earlier times, a significant trend of the increasing VBT utilization rate was observed. Furthermore, survival analyses demonstrated no significant differences in OS, DFS, DMFS, and LRFS between EBRT ± VBT and VBT, regardless of HIR group. Additionally, for patients who met the GOG-249 HIR criteria and did not receive pelvic lymphadenectomy, EBRT showed higher DFS and LRFS than VBT alone (*p* < 0.05).

Based on the real-world data, the present results showed the proportion of patients who received EBRT ± VBT in the whole cohort, GOG-249 cohort, PORTEC-2 cohort, and ESMO-ESGO-ESTRO HIR cohort were 87.5%, 87.5%, 100%, and 100%, respectively, in 2005. While the proportion of EBRT utilization declined to 50%, 62.5%, 37.5%, and 50%, respectively, in 2015. Co-occurring with the decline of EBRT utilization, a significantly increasing trend in the utilization of VBT among the entire cohort and HIR groups was observed between 2000 and 2015 by the Man-Kendall test (*p* < 0.05). Similar RT use trends have also been reported in America using the National Cancer Database (NCDB). Parth A. Chodavadia et al. evaluated the off-study use of RT and found the largest increase in VBT + CT use in 2009 and in the year GOG-249 was initiated [13]. Sara J. Zakem et al. demonstrated that the utilization rate of VBT alone was higher than that of pelvic EBRT. The use of VBT increased from 21.5% to 30.3% between 2010 and 2015 [14]. It can be observed that the timing of announcements of clinical trials and guidelines potentially impacts the adjuvant treatment options. The GOG-249 was launched in 2009 and published its latest research results in 2017, which compared VBT and chemotherapy (VCB/C) with EBRT in HIR patients and HR patients. The latest results showed no significant differences in 5-year RFS and OS between EBRT and VCB/C. However, the proportion of acute toxicity in the VCB/C group was higher. VCB/C is not superior to EBRT, and EBRT remains an effective and appropriate primary adjuvant therapy for patients with HIR in endometrial cancer [15]. GOG-249 enrolled patients with higher risk factors than those in the PORTEC and ESMO-ESGO-ESTRO HIR groups, including those with G3 and deep MMI. Thus, the rate of EBRT utilization in the GOG-249 group was higher than in the other two HIR groups, and VBT alone was considered insufficient. However, GOG-249 confirmed that VBT plus chemotherapy was not superior to EBRT, regardless of RFS or OS but had higher toxicities with the addition of chemotherapy. The effect of adjuvant chemotherapy has not been previously demonstrated and is listed as the category 2B recommendation in the NCCN guideline. In the present study, the rate of adjuvant chemotherapy was rather low (20.9% in the whole cohort).

Additionally, PORTEC-2, released in 2002, compared the efficacy and toxicity of EBRT with VBT on treatment de-escalation in HIR. The hypothesis that VBT was equally efficient as EBRT with lower radiation-induced toxicities may have influenced radiotherapy patterns in practice, although the final results had not yet been published [10]. In our study, the utilization of VBT alone had experienced a significant increase since around 2003 and the most considerable growth around 2005. The final results of PORTEC-2, published in 2008, demonstrated that VBT had a similar rate of 5-year vaginal recurrence as EBRT (1.8% vs. 1.6%, *p* = 0.74) and a similar OS rate [10]. The long-term results of PORTEC-2. published in 2018. demonstrated that the VBT group had a higher pelvic recurrence rate (6.3% vs. 0.9%, *p* = 0.004) [16]. It suggests that we should be cautious about using hypothetical conclusions of clinical trials in practice before the long-term results are published. Notably, in the subsequent review of the PORTEC-2 study, it was found that the tumor grade, i.e., patients with G1 grade, increased from 48% to 79% after review, and the depth of myometrial invasion may have also been overestimated. So, the PORTEC-2 study may not be an accurate representation of HIR patients, and the critical prognostic features should be centrally reviewed by pathologists [17]. As for long-term survival, 10-year OS was 69.5% in the VBT group and 67.6% in the EBRT group, with no significant difference (*p* = 0.72) [4]. Likewise, Peter J. Zavitsanos et al. examined the RT modalities in stage I ECs who met the HIR criterion of PORTEC-2 in the National Cancer Institute’s Surveillance, Epidemiology, and End Results database (SEER), and they found that the rate of VBT increased and EBRT decreased from 2004 to 2011 [18]. The results from PORTEC-2 corroborated that VBT alone could be regarded as the standard adjuvant radiation modality for HIR endometrial cancer. However, despite the evidence from randomized trials, a study from the United States showed that 9% of HIR ECs received EBRT [19].

The OS for early-stage EC is relatively satisfactory, with a 5-year OS of more than 90%. However, OS may not be the only endpoint for early-stage EC since there are many other causes of death [20]. Compared to VBT alone, pelvic EBRT can further reduce locoregional recurrence by irradiating areas of micrometastasis and subclinical lesions [21]. Notably, the increase in local control only translated into a survival benefit in a small group for early-stage EC [21,22]. Patients need to be more carefully selected for EBRT, as the cost and toxicity of EBRT are considered higher than VBT alone [12]. The subgroup analysis in our study revealed that EBRT showed better DFS and LRFS than VBT alone for patients who did not undergo lymphadenectomy after PSM in the GOG-249 HIR group. Lymphadenectomy may remove cancer cells that have spread to nearby lymph nodes [23]. EBRT can damage subclinical occult tumors in the pelvis to improve efficacy. Considering the differences between the other two criteria, the GOG-249 patients had higher risk characteristics, so our subsequent study excluded the patients with FIGO II, and the results indicated that the EBRT group had longer DFS than the VBT group in patients who did not undergo lymphadenectomy (88.1% vs. 62.3%, *p* = 0.025). This finding was consistent with previous research [21,24]. Junzo P Chino et al. demonstrated that EBRT was associated with increased survival compared to VBT alone in the high-risk Stage I patients who did not receive lymph node dissection (*p* = 0.01) [24]. Besides, the thoroughness of the lymphadenectomy was also associated with survival [25]. Removal of more than 10 lymph nodes conferred further improvement in survival [26].

However, controversy remains regarding the definition of adequate lymphadenectomy and the indications for lymphadenectomy, especially in early-stage EC patients [27,28]. SLN dissection, which usually removes less than 10 lymph nodes, may represent an alternative to full dissection in selected lower-risk patients [5]. However, the role of SLN is in the experimental stage [29]. Thus, for HIR patients who do not choose lymphadenectomy during surgery, adjuvant EBRT is recommended [19,30]. In addition, other clinicopathologic indicators would be helpful for further risk classification and identifying patients who may benefit from EBRT. Being older than 65 years old was considered a poor independent factor in early-stage EC patients [31], and age-related factors were considered by GOG-249 and PORTEC-2 but not by ESMO-ESGO-ESTRO. The impact of age on survival may be multifactorial since patients with older age may not be administrated with lymphadenectomy, pelvic EBRT, and systemic therapy, which is due to it being recommended less by their physicians [32]. However, some investigators proposed that advanced age should not be considered a reason not to perform optimal treatment, and pelvic EBRT for patients older than 70 is feasible and well tolerated [33].

In addition to survival benefits, economic costs and toxicities need to be considered when making decisions about adjuvant radiotherapy. In the present study, patients who received EBRT experienced a significantly higher rate of acute toxicities than VBT alone (*p* < 0.05). On the other hand, EBRT ± VBT tends to result in more extended hospitalizations, higher charges, and higher toxicity rates. Some cost-effectiveness studies demonstrated that VBT alone is the most cost-effective adjuvant regimen with the highest 5-year quality-adjusted survival rate (86%) compared to EBRT ± VBT [34].

A significant strength of our study is that we enrolled a relatively large patient population from multi-institutions. Furthermore, we revealed for the first time the trends in RT utilization for HIR early-stage EC patients in China. There were still some limitations in the present study. Our study provided evidence for radiotherapy optimization in patients with HIR. Still, we did not identify a specific patient subset who would benefit from EBRT based on the existing risk classifications. The patient characteristics in the present study were different from those of patients enrolled in the GOG-249 or PORTEC-2 cohorts. Still, our results may genuinely represent real-world clinical practice for HIR patients in China. Although it is a retrospective study, our study can be regarded as a supplement to the current treatment decision-making system and provide evidence for prospective studies. Recently, studies on molecular classification to guide adjuvant treatment have been prospectively initiated.

## 5. Conclusions

EBRT did not significantly improve OS, DFS, LRFS, and DMFS in the HIR patient group defined by GOG-249, PORTEC 2, or ESMO-ESGO-ESTRO. Thus, VBT could be regarded as a standard adjuvant radiation modality for patients with HIR. EBRT should be administered to selected HIR patients who did not receive lymphadenectomy.

## Figures and Tables

**Figure 1 cancers-14-05129-f001:**
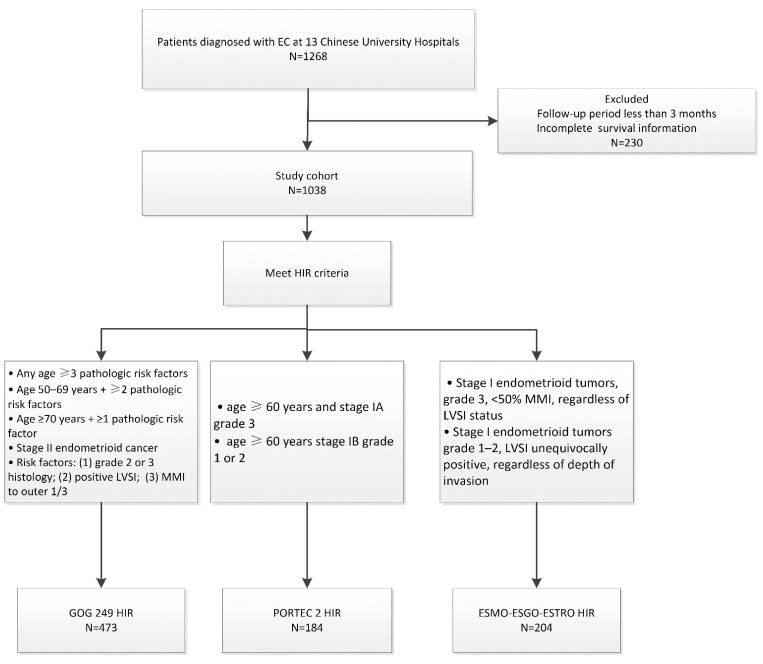
Exclusion criteria and patient cohort selection diagram.

**Figure 2 cancers-14-05129-f002:**
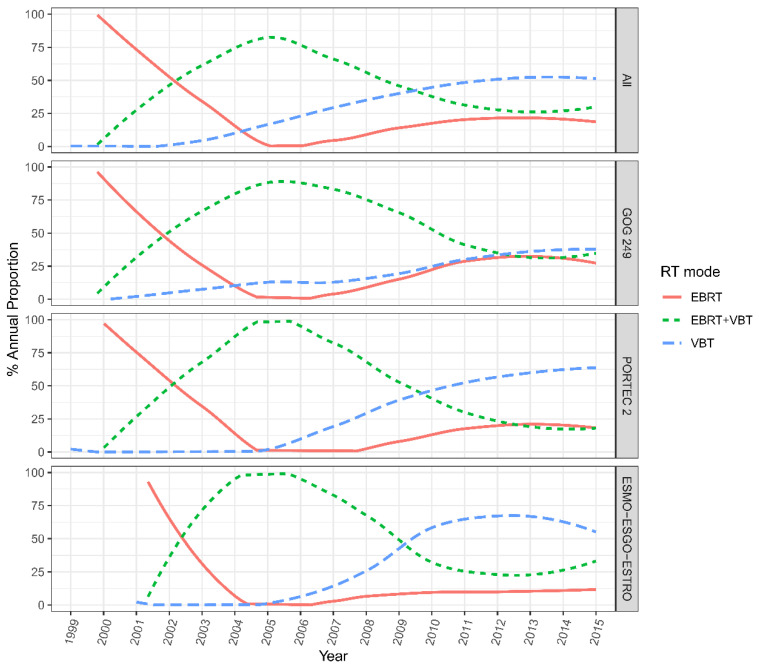
Evolving trends in RT mode utilization in patients meeting HIR criteria across the 13 medical centers in China. EBRT = external beam radiation therapy; VBT = vaginal brachytherapy.

**Figure 3 cancers-14-05129-f003:**
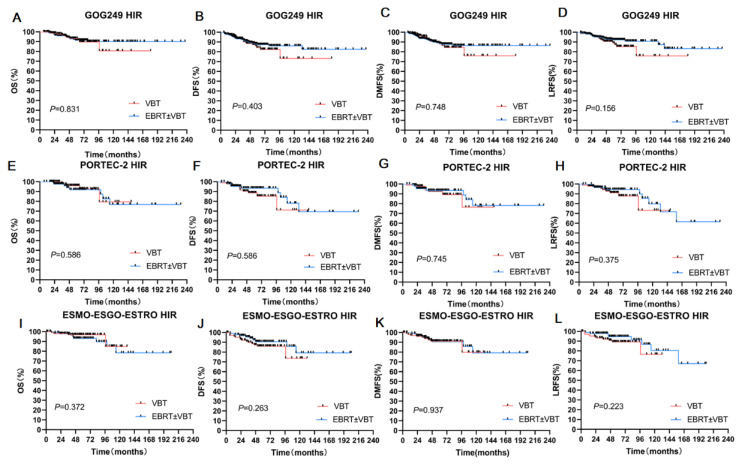
Subgroup analyses of three HIR groups. (**A**) OS for two RT modes in the GOG-249 HIR Group. (**B**) DFS for two RT modes in the GOG -249 HIR group. (**C**) DMFS for two RT modes in the GOG -249 HIR group. (**D**) LRFS for two RT modes in the GOG-249 HIR group. (**E**) OS for two RT modes in the PORTEC -2 HIR group. (**F**) DFS for two RT modes in the PORTEC -2 HIR group. (**G**) DMFS for two RT modes in the PORTEC -2 HIR group. (**H**) LRFS for two RT modes in the PORTEC -2 HIR group. (**I**) OS for two RT modes in the ESMO-ESGO-ESTRO group. (**J**) DFS for two RT modes in the ESGO-ESTRO HIR group. (**K**) DMFS for two RT modes in the ESGO-ESTRO HIR group. (**L**) LRFS for two RT modes in the ESGO-ESTRO HIR group.

**Figure 4 cancers-14-05129-f004:**
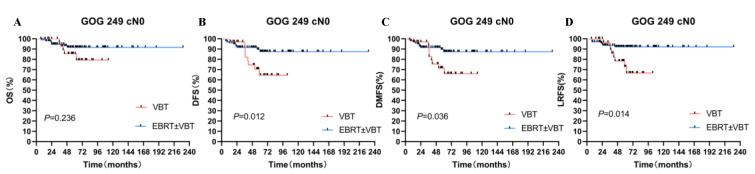
Subgroup analysis in the GOG249 HIR cohort. (**A**) OS for two RT modes. (**B**) DFS for two RT modes. (**C**) DMFS for two RT modes. (**D**) LRFS for two RT modes.

**Table 1 cancers-14-05129-t001:** Baseline characteristics of patients in the three HIR cohorts.

	GOG-249 (*n* = 473)	PORTEC-2 (*n* = 184)	ESMO-ESGO-ESTRO (*n* = 204)
Time periods			
1999–2005	30 (6.3%)	16 (8.7%)	10 (4.9%)
2005–2010	70 (14.8%)	27 (14.7%)	33 (16.2%)
2010–2015	373 (78.9%)	141 (76.6%)	161 (78.9%)
Age			
≤60 years	309 (65.3%)	0 (0%)	143 (70.1%)
>60 years	164 (34.7%)	184 (100%)	61 (29.9%)
FIGO stage			
IA	59 (12.5%)	29 (15.8%)	159 (77.9%)
IB	279 (60.0%)	155 (84.2%)	45 (22.1%)
II	135 (28.5%)	0 (0%)	0 (0%)
Lymphovascular invasion			
no	332 (70.2%)	163 (88.6%)	92 (45.1%)
yes	141 (29.8%)	21 (11.4%)	112 (54.9%)
Histologic grade			
Grade 1	66 (14.0%)	69 (37.5%)	30 (14.7%)
Grade 2	278 (58.8%)	84 (45.7%)	68 (33.3%)
Grade 3	129 (27.2%)	31 (16.8%)	106 (52%)
Radiation therapy modality			
VBT	147 (31.1%)	95 (51.6%)	102 (50%)
EBRT ± VBT	326 (68.9%)	89 (48.4%)	102 (50%)
Chemotherapy			
yes	100 (21.1%)	24 (13%)	53 (26%)
no	335 (70.8%)	140 (76.1%)	133 (65.2%)
missing	38 (8.0%)	20 (10.9%)	18 (8.8%)
Received lymph node dissection			
yes(pN0)	339 (71.7%)	119 (64.5%)	155 (76%)
Full dissection	303 (89.4%)	106 (89.1%)	134 (86.5%)
SLN	36 (10.6%)	13 (10.9%)	21 (13.5%)
no(cN0)	134 (28.3%)	65 (35.3%)	49 (24.0%)

**Table 2 cancers-14-05129-t002:** Survival analysis for patients who received VBT or EBRT ± VBT in the three HIR groups.

Cohorts		RT Mode	5-Year OS	5-Year DFS	5-Year DMFS	5-Year LRFS
GOG-249	Full cohort	VBT	93.30%	84.20%	87.90%	89.00%
		EBRT ± VBT	91.50%	87.40%	87.70%	92.20%
		*P*	0.831	0.403	0.748	0.156
	After matching	VBT	93.80%	84.50%	88.40%	89.40%
		EBRT ± VBT	91.90%	84.60%	85.30%	91.20%
		*P*	0.855	0.834	0.855	0.311
PORTEC-2	Full cohort	VBT	96.00%	93.50%	92.50%	91.10%
		EBRT ± VBT	91.90%	88.70%	93.50%	94.50%
		*P*	0.586	0.331	0.745	0.375
	After matching	VBT	96.00%	88.70%	92.50%	91.10%
		EBRT ± VBT	96.40%	95.10%	95.10%	97.10%
		*P*	0.976	0.186	0.462	0.174
ESMO-ESGO-ESTRO	Full cohort	VBT	96.90%	87.40%	90.60%	89.20%
		EBRT ± VBT	93.10%	90.60%	91.20%	94.60%
		*P*	0.372	0.263	0.937	0.223
	After matching	VBT	96.90%	87.40%	91.20%	89.20%
		EBRT ± VBT	94.30%	91.10%	91.00%	95.90%
		*P*	0.489	0.233	0.850	0.170

## Data Availability

All data analyzed during this study are included in the published article.

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
