# Peer review of "Utilization Trend and Comparison of Different Radiotherapy Modes for Patients with Early-Stage High-Intermediate-Risk Endometrial Cancer: A Real-World, Multi-Institutional Study"

_cancers, 2022, doi:10.3390/cancers14205129_

Round 1

Reviewer 1 Report

The authors have well conducted and compared the different radiotherapy treatments for endometrial cancer patients. The results of the survey are well documented in graphs and also tabulated. In almost all cases, VBT + EBRT shows a better survival outcome than VBT alone. Why do authors conclude that VBT is a standard treatment option rather than combination therapy? Acronyms need to be defined.

Author Response

Thank you very much for the pertinent advice regarding our article. We evaluated the survival outcomes of HIR groups meeting GOG-249, PORTEC 2, and ESMO-ESGO-ESTRO criteria. From the survival curves perspective, the combined VBT+EBRT appeared to have better survival outcomes with no statistical significance, even after balancing other clinicopathological factors using Propensity Score Matching. Besides, the further 5-year survival analyses confirmed no significant differences between VBT alone and VBT + EBRT groups. The two RT modes had similar long-term survival outcomes and local recurrences, but pelvic EBRT was associated with more prolonged treatment and higher medical costs. Also, pelvic EBRT had a higher rate of acute and late toxicities than VBT alone, which was reconfirmed by our study (Table S2). Based on our analysis, it can be concluded that VBT alone was not inferior to EBRT in early-stage ECs with HIR. Thus, we were confident to conclude that vaginal brachytherapy(VBT)alone can be regarded as a standard adjuvant radiation option for HIR patients with similar outcomes and lower toxicities. And we found EBRT could improve survival only in selected patients group. According to your advice, we have revised expressions in the simple summary and abstract to better interpret our conclusions:

Simple summary (page 1, lines 40-43): It confirmed that pelvic external beam radiation therapy (EBRT) showed the survival advantage over vaginal brachytherapy (VBT) alone only in selected patients with HIR.

Abstract (page 2, lines 210-212): EBRT should be administrated to selected HIR patients who meet the GOOG-249 criterion and did not receive lymph node dissection.

Also, we modified the part of the Conclusion for better presentation of the result (page 11, line 568-569): EBRT should be administered to selected HIR patients who did not receive lymphadenectomy.

And the acronyms firstly appeared in the abstract were explained in page 2(lines 37-43).

Reviewer 2 Report

In the presented study, the authors conducted a retrospective analysis of 1268 patients with early endometrial cancer from 13 Chinese medical institutions. Analyzed group of the patients were reclassified into HIR according to the a little bit different protocols GOG-249, PORTEC-2 and ESMO-ESGO-ESTRO Consensus Conference on Endometrial Cancer (2016). Regardless of the adopted criteria for including the groups in to HIR the analysis carried out allowed for the assessment of changes in the selection of the adjuvant radiotherapy methods that took place in China in the analysed period. The rate of VBT alone increased to 50% in 2015, while the rate of EBRT+VBT utilization declined to around 25%. Moreover, the authors noted the lack of the differences in OS, DFS, LRFS, DMFS between VBT alone and EBRT+/- VBT. Similar conclusions are known  from previous studies and consensus of international scientific societies, although an advantage is undoubtedly a fairly large group of patients, retrospectively analyzed on the basis of classic prognostic factors included in the histopathological report. The subgroup deserves attention is subgroup analyses in the GOG-294 HIR cohort: the authors showed that for the patients without lymph node dissection, the DFS, DMFS and LRFS of patients who were treated with EBRT+/- VBT were significant better than for patients who received VBT only. This fits in with the discussion of the importance of surgical assessment of lymph nodes in patients with massive LVSI, G3 and to what extent imaging tests can replace this assessment. In the section 3.5 the authors mention the role of adequate lymphadenectomy in the analyzed group of patients. While the influence of the lack of lymphadenectomy seems to be clear, the mention of influence of adequate lymphadenectomy requires more comment. What is its definition, how many patients did not meet this requirement . The survival curves of these patients in comparison with adequate lymphadenectomy would be interesting. The identification of HIR patients operated for early endometrial cancer who will benefit most from EBRT +/- VBT remains open. Molecular profiling of the endometrial cancer undoubtedly facilitate this.           

Author Response

Thank you for your positive and constructive comments on our article. We illustrated the definition of adequate lymphadenectomy and added it in the part of Method, page 3(line 292-297): Patients who only received preoperative imaging assessment of lymph nodes without lymphadenectomy staging would be classified into the cN0 group, which was considered inadequate lymph node assessment. Patients who undergo lymphadenectomy or suspicious lymph node biopsy would be categorized into the pN0 group after pathological confirmation of negative, which was considered to be an adequate assessment.

We also modified the part of Result for simplicity accordingly (page 9, line 413 and 416): For the cN0 patients’ group, the 5-year DFS, 5-year DMFS, and 5-year LRFS of patients...; While for the pN0 patients’ group, there were no significant differences in...;

In our study, lymphadenectomy included pelvic with or without para-aortic lymph node dissection, and the count of lymph nodes removed was usually more than 10. Sentinel lymph node (SLN) dissection was defined as another mode of lymphadenectomy, usually with a dissection of less than ten lymph nodes. Both two methods were considered to be adequate lymph node assessments. We added a table to illustrate the status of lymphadenectomy in the present study (Table R1 and Table 1-Revised in the manuscript). Table R1 showed that only a small proportion of patients received SLN, so a further comparison between comprehensive lymphadenectomy and SLN was not conducted. Then, we investigated the survival outcomes between patients without lymphadenectomy and patients with adequate lymph node assessment and added these results as follows (Figure R1). Survival curves showed no statistical significance between the two patients’ groups even after balancing other clinicopathological such as age, FIGO stage, MMI, LVSI, and radiotherapy modality factors propensity score matching. It can be concluded that lymphadenectomy status may not be the independent survival factor. Previous studies shown dissection of more than ten lymph nodes were associated with improved survival[1], but only 18% received pelvic EBRT in their research. Long-term survival and recurrence were multi-factorially influenced by therapy-related factors. We added in the Discussion part to better illustrate the role of lymphadenectomy, page11 (Lines 533-543 ): Junzo P Chino et al. demonstrated that EBRT was associated with increased survival than VBT alone in the high-risk patients with Stage I who did not receive lymph node dissection (p=0.01)[24]. Besides, the thoroughness of lymphadenectomy was associated with survival, too[25]. Removal of more than ten lymph nodes conferred further improvement  in survival[26].

However, controversy remains regarding the definition of adequate lymphadenectomy and the indications of lymphadenectomy, especially in early-stage ECs[27,28]. SLN dissection, which usually removes less than ten lymph nodes, may represent an alternative to full dissection in selected lower-risk patients[29]. But the role of SLN is in the experimental stage[30]. Thus, for HIR patients who do not choose lymphadenectomy during surgery, adjuvant EBRT is recommended[19,31].

We agree with your opinion that future molecular profiling would give answers to the optimized adjuvant treatment strategy. We aim to address these questions by recapitulating the experiences from retrospective data and conducting prospective trials.

Please see the Tables and Figure in the attachment
